# Entrepreneurial Intention Among Engineering Students: The Moderating Role of Entrepreneurship Education in Japan

**DOI:** 10.3390/bs15050663

**Published:** 2025-05-12

**Authors:** Karin Kurata, Kota Kodama, Itsuki Kageyama, Yoshiyuki Kobayashi, Yeongjoo Lim

**Affiliations:** 1Graduate School of Design and Architecture, Nagoya City University, 1-1-10, Kita Chikusa, Chikusa-Ku, Nagoya 464-0083, Japan; c235802@ed.nagoya-cu.ac.jp; 2School of Pharmacy and Pharmaceutical Sciences, Hoshi University, 2-4-41 Ebara, Shinagawa-Ku, Tokyo 142-8501, Japan; kageyama.itsuki@hoshi.ac.jp (I.K.); kobayashi.yoshiyuki@hoshi.ac.jp (Y.K.); 3College of Business Administration, Ritsumeikan University, 2-150 Iwakura-Cho, Ibaraki 567-8570, Japan

**Keywords:** entrepreneurship education, entrepreneurship intention, theory of planned behavior

## Abstract

With the growing interest in entrepreneurship, increased attention has been paid to entrepreneurship education. In recent years, attention has been paid to entrepreneurship education in engineering majors. This study examines the impact of attitude, subjective norms, and self-efficacy toward entrepreneurial intention and the moderating role of entrepreneurship education among students at the National Institute of Technology, Tsuruoka College, Japan. This study was grounded in the theory of planned behavior and social cognitive theory, suggesting a new approach to examining entrepreneurship education in engineering students. A questionnaire survey was conducted with 275 Japanese students (150 second-year students and 125 fourth-year students). Employing structural equation modeling, the findings indicated that attitude and self-efficacy significantly influence entrepreneurship intentions, with notable differences between the second- and fourth-year students. Results have suggested focusing both on project-based learning and theory-based learning to nurture knowledge, skill, and mindset to comprehensively develop entrepreneurial intention.

## 1. Introduction

Entrepreneurship is the process of identifying, co-creating, and evaluating opportunities to develop new value ([43]). Recently, the interest in the entrepreneurship research field has been focused on entrepreneurship education. Entrepreneurship education is defined as the acquisition of skills essential for entrepreneurs, such as leadership, creativity, risk management, and perseverance ([58]). With the realization that technology-driven entrepreneurship can mitigate regional economic challenges, entrepreneurship education research has expanded from the business field towards new fields including the science and engineering fields ([11]; [36]; [51]). In the Global Entrepreneurship Monitor survey in 2023, Japan ranked lowest among the participating countries in terms of the knowledge and skills required to start a business ([21]). To address this issue, various Japanese educational institutions, including middle schools ([62]), high schools ([59]), and universities ([38]; [57]), have initiated efforts to cultivate entrepreneurship intentions through education. However, to the best of author’s knowledge, there are few studies which focused on engineering and science major students. This is crucial because the effectiveness of entrepreneurship education can vary according to the academic discipline ([41]). This study focuses on the engineering students at the National Institute of Technology (NIT), Tsuroka College, Japan, to explore the factors that influence students’ entrepreneurship intentions. NIT supports students in becoming engineers while providing liberal arts and engineering majors and subjects ([20]). It was established during a period of high economic growth to satisfy the demands of engineers in the manufacturing sector. Currently, approximately 10,000 students are enrolled annually in 58 Japanese schools. The characteristics of NIT lie in its 5-year integrated education program, which allows students to start their research activities as early as 15 years of age. Moreover, NIT students are expected to gain experience developing goods and services that show a deep connection with entrepreneurship. Students at NIT are expected to support economic development as most engineering major students in the world, and it is important to identify the factors influencing entrepreneurial intention. The current study is expected to shed light on the entrepreneurship education for youth in the field of engineering. Moreover, this research results will support practitioners, researchers, and policy makers who aims to foster entrepreneurship education. The following literature review will be delivered to explain the gap between previous studies and the current study. Then, the hypothesis development will be presented to support our hypotheses. Finally, the methodology, results, and discussion will be explained to share our findings and improvements plan.

Previously, studies have examined the youth in engineering majors regarding entrepreneurship and entrepreneurship education. For instance, [26] ([26]) studied 384 high schools offering vocational training in Brazil. The analysis used PLS-SEM to identify the effects of five abilities (sociability, planning, leadership, innovation, and risk-taking) on entrepreneurial intention by mediating self-efficacy. The results showed that all five skills developed entrepreneurial intention by mediating self-efficacy. To the best of the author’s knowledge, this is the first study to conduct a study of high school students in vocational schools. However, because this study did not describe entrepreneurship education, it was difficult to identify its effects. This highlights the need to conduct studies to identify the role of entrepreneurship education.

[44] ([44]) conducted a study on students in Technical and Vocational Education and Training in Ghana. They administered questionnaires based on the theory of planned behavior to 376 female students. The results showed the effect of entrepreneurship education on entrepreneurship intention. Moreover, the results showed that students’ friends and family’s emotional, social, economic, and political support had a greater influence on entrepreneurship intention than entrepreneurship education. A limitation of this study is that it only conducted a questionnaire survey after the students received the entrepreneurship education. More specifically, this study did not identify the effects of entrepreneurship education by comparing the differences before and after the entrepreneurship education, making it difficult to specify the changes generated by entrepreneurship education. This study suggests the need to conduct questionnaire surveys before and after students receive entrepreneurship education to identify the role of entrepreneurship education.

[11] ([11]) conducted a study with 423 students majoring in production and computer engineering in Spain. This study identifies how three aspects of entrepreneurship motivation—achievement, independence, and economic independence—influence entrepreneurship intention. Moreover, it examines how entrepreneurship education can moderate these three factors to influence entrepreneurship intentions. By conducting both pre- and post-questionnaire surveys, this study found that the need for independence and economic independence had positive and significant effects on entrepreneurial intentions. Moreover, they found no moderating effect on entrepreneurship education.

[3] ([3]) conducted a study with 448 engineering students in Pakistan. This study has analyzed how attitude, subjective-norm, and perceived behavioral control will influence on the entrepreneurial behavior while mediating the entrepreneurial intention. Moreover, the moderating role of entrepreneurial motivation was analyzed. Results found that both attitude and perceived behavioral control have positive and significant influence on the entrepreneurship intention. However, the subjective norm did not significantly influence the entrepreneurship intention. Finally, the mediating role of entrepreneurial intention was found to be significant for attitude and perceived behavior control with entrepreneurial behavior. The course objectives and materials for entrepreneurship education was not clarified in the paper, making it challenging to compare the results with other research papers.

The above literature review found that studies on entrepreneurship education have both significant and non-significant findings on the development of entrepreneurship intention. This shows that the question of whether entrepreneurship education develops entrepreneurial intentions is controversial. Moreover, studies in the engineering field focused on practical implications, and little attention was paid to theoretical implications ([41]; [47]). Based on this discussion above, the objective of this study is to conduct a quantitative analysis on the engineering major students while identifying the role of entrepreneurship education. We will conduct analysis and discussion while combining theory of planned behavior and social cognitive theory. Specifically, we will identify how attitude, subjective norms, and self-efficacy will influence entrepreneurship intention. Also, how entrepreneurship education will moderate each relationship. The novelty of this study is that current study has adopted a new methodology to analyze the effect of entrepreneurship education by comparing the data between second- and fourth-year students while conducting a questionnaire survey to 310 students at the same time. This has minimized the risk of gathering the exaggerated results on the effect of attitude, subjective norm, and self-efficacy on entrepreneurship intention. Moreover, to conduct a study based on the theory of planned behavior while improving this theory by changing the perceived behavioral control to self-efficacy while combining the social cognitive theory, this study has created a new pathway to analyze the entrepreneurship education for engineering major students.

## 2. Hypothesis Development

The theory of planned behavior is a theory which predicts a person’s intention and behavior through attitude, subjective norms, and perceived behavior control toward a certain action ([1]; [36]; [53]). Moreover, it is known that when predicting intention and behavior using the theory of planned behavior, it is most effective when the event is rare, difficult to observe, or involves an unpredictable time lag ([32]). As entrepreneurship requires extensive planning to identify and solve problems ([32]), the theory of planned behavior is considered appropriate for examining the development process of entrepreneurship intention. Moreover, the theory of planned behavior is commonly used to clarify the effect of entrepreneurship education on entrepreneurship intention ([7]; [37]; [47]; [49]; [53]). Further, meta-analyses have suggested that entrepreneurship education studies have utilized attitude, subjective norms, and perceived behavioral control as the most frequently used variables in promoting entrepreneurial intentions ([50]).

Attitude refers to people’s favorable or unfavorable opinions toward something ([1]; [36]). Favorable behavior creates a greater tendency toward intentions and behavior. Subjective norms refer to the perceptions of the people around us formed by social reference groups, such as friends, classmates, and family, to direct our intentions and behaviors. The favorable opinion of the reference group toward a certain action encourages people to generate intentions and behaviors. In this study, we adopted self-efficacy as a substitute for perceived behavioral control. Self-efficacy is a concept developed by [10] ([10]) in social cognitive theory ([9]). Self-efficacy predicts the individual’s actions regardless of the existence of alternatives, the time it takes to act, perseverance toward barriers, and the presence or absence of opportunities to act ([10]; [45]; [61]). However, the perceived behavioral control clarifies the person’s perception of how easy or difficult it is to carry out the behavior. As perceived behavioral control reflects past experiences and anticipated barriers, recent research on entrepreneurship education has adopted self-efficacy ([23]; [63]); this study decided to use self-efficacy as it is more appropriate in the research model. Intention represents how much effort a person will put into performing a certain behavior ([23]). A study found that entrepreneurship education influences both entrepreneurship intention ([2]) and entrepreneurship behavior ([6]; [7]). Considering the time to initiate entrepreneurial action, this study has decided that it is appropriate to investigate entrepreneurship intention rather than entrepreneurship behavior.

### 2.1. Attitude, Subjective Norms, and Self-Efficacy Toward Entrepreneurship Intention

The theory of planned behavior signifies that a person’s future actions are preceded by intention; the stronger the intention for a certain action, the more likely that the behavior will be executed ([36]). Studies on entrepreneurship education based on the theory of planned behavior revealed that attitude ([49]; [56]), subjective norms ([37]), and self-efficacy ([23]; [63]) have positive and significant effects on entrepreneurial intention. Based on the above discussion, we developed the following hypothesis (Figure 1).

**H1(a).** 
*Attitude toward entrepreneurship positively and significantly affects entrepreneurial intentions.*


**H1(b).** 
*Subjective norms toward entrepreneurship positively and significantly affect entrepreneurial intention.*


**H1(c).** 
*Self-efficacy toward entrepreneurship positively and significantly affects entrepreneurial intentions.*


### 2.2. Impact of Entrepreneurship Education Toward Entrepreneurial Intentions

Studies have conducted pre- and post-questionnaire surveys on entrepreneurship education for students to identify the role of entrepreneurship education ([2]; [16]; [13], [14]; [28]; [29]; [35]). However, some have suggested that conducting a questionnaire survey immediately after class could deliver a higher degree of effect than the actual effect ([47]). Therefore, when identifying the effects of education, it is essential to let a certain amount of time pass by after the course ([15]). A few studies on entrepreneurship education that were considered in the above discussion ([4]; [15]; [47]). As cross-sectional studies have been the norm in recent year ([4]), this study conducted a questionnaire survey to second- and fourth-year students while fourth-year students received an entrepreneurship education during their third year.

Studies have found that entrepreneurship education positively and significantly influences entrepreneurship intention among students ([23]; [40]; [49]). Entrepreneurship education improves students’ attitude by explaining the benefits of entrepreneurship favorably ([36]). Similarly, self-efficacy is developed by repeatedly presenting solutions to problems through group work in class, increasing students’ confidence in their abilities ([23]; [60]). Based on the discussion above, we developed the following hypothesis.

**H2(a).** 
*Entrepreneurship education positively and significantly influences the relationship between attitude and entrepreneurial intention.*


**H2(c).** 
*Entrepreneurship education positively and significantly influences the relationship between self-efficacy and entrepreneurship intention.*


Studies have shown that entrepreneurship education has a negative and significant effect on the relationship between subjective norms and entrepreneurial intention ([36]). Entrepreneurship education helps students acquire the ability to make their own decisions without relying on others’ opinions of others ([30]). Therefore, when students undertake entrepreneurship education, the degree to which subjective norms influence entrepreneurship intentions decreases. Based on the above discussion, we developed the following hypotheses (Figure 1).

**H3(b).** 
*Entrepreneurial education negatively and significantly influences the relationship between subjective norms and entrepreneurship intention.*


## 3. Methodology

### 3.1. Participants

This study conducted a questionnaire survey with in total 310 students (158 s year students and 152 fourth year students) at the NIT, Tsuruoka College, both questionnaire survey was conducted in September 2023 using an online questionnaire survey. The survey was conducted as part of the class; all four courses in two grades were encouraged to participate in the survey. We collected 275 valid responses (150 second- and 125 fourth-year students), with a valid response rate of 89% (Table 1). The total number of valid responses is 275, while the population size is 310. There was a 1.95% sampling error rate for a 95% confidence level. Moreover, the error rate was low enough to be appropriate for a statistical study ([12]). For data acquisition, to provide a minimum knowledge of entrepreneurship before conducting the questionnaire survey, we held a five-minute presentation on the general idea of entrepreneurship. The information on participants’ consent was received before the questionnaire survey. Moreover, it was clearly explained to the students that the questionnaire survey was only used for the research purposes, all of the information would be gathered and utilized anonymously, the questionnaire survey was not mandatory, and the data would not affect the grades.

### 3.2. Constructs

This study developed a questionnaire based on previously published articles to measure the degree of attitude, subjective norms, self-efficacy, and entrepreneurial intention. We adopted the scales developed by [34] ([34]) for attitude, subjective norms, and entrepreneurship intention. To measure subjective norms, the following question was modified: “If you decided to create a firm, would people in your close environment approve of that decision?”. There were four types of perspective in the question, including the close environment, family, friends, and colleagues. We changed the types of perspective to that of family, friends, classmates, and teachers to meet the environment of students. For self-efficacy, we adopted the scale developed by [17] ([17]). We divided two questions into four questions, because each question contained two different objectives. The questions include “I show great aptitude for creativity and innovation” and “I show great aptitude for leadership and problem-solving”. Moreover, we deleted one question that included the concept of entrepreneurial behavior, as it was not appropriate for our research objective. The deleted question was: “I can develop and maintain favorable relationships with potential investors”. For these four variables above, participants were asked to respond using a five-point Likert scale ranging from 1 = strongly disagree to 5 = strongly agree.

### 3.3. Data Analysis

We have conducted a covariance-based structural equation modeling (hereafter SEM) to test our hypotheses using SPSS 25 and AMOS. SEM is a statistical methodology to build model by developing variables to test the relationships between them ([31]). The goal of SEM is to test hypotheses through causal relationships while combining confirmatory factor analysis (CFA) and path analysis ([48]). Partial least squares structural equation modeling (PLS-SEM) is similar to SEM, while both statistical methods identify the relationship between each variable. However, as previous study mentioned that SEM is designed to test the established theoretical frameworks including measurement and structural models. On the other hand, PLS-SEM is designed to identify the user’s structural model objective is to predict and explain the target outcome as obtained by the in-sample and out-of-sample metrics ([22]). Therefore, this study is decided to utilize the SEM for the statistical methodology. In the descriptive analysis, we performed a correlation analysis to identify the relationship between variables. The coefficients between 0 to ±0.3 show a low correlation. On the other hand, coefficients between ±0.3 to ±0.7 indicate a medium correlation, and coefficients between ±0.7 to ±1 suggest a high correlation ([8]).

First, CFA was conducted to determine the relationship between each theoretical variable while assessing the model fit of the empirical data using two methods, including absolute fit indices and incremental fit indices ([48]). Absolute fit indices identify how well a model fits the sample data by comparing multiple proposed models to identify the better fit ([24]; [39]). Incremental fit indices, also known as relative fit indices, compare the chi-square value to a base model, with its null hypothesis being that all variables are uncorrelated. For current study, we have selected absolute fit indices as CMIN/DF and RMSEA. As for the incremental fit indices, we have selected IFI and CFI. Moreover, we verified the reliability of the model through convergent validity. Convergent validity is to assess the accuracy of elements and latent variables ([48]). The criterion for convergent validity is to assess whether the composite reliability (CR) value is 0.7 or above ([19]). Finally, to identify the explanatory power of the regression analysis, we calculated the R-square value.

Second, we conducted a pass analysis to identify the relationship between independent and dependent variables. Moreover, to identify the moderating role of entrepreneurship education, we conducted a multi-group analysis. Similarly, there is a method called moderated regression analysis to identify the moderator’s effect on the relationship between the independent variable and dependent variables. As this study is to compare the different age groups with its aim of identifying the different effects of independent variables on the dependent variables, the current study has selected multi-group analysis. In the beginning, the data were divided into second and fourth-year response data. Then, to test the statistical significance of the moderating effect, we have applied a chi-square difference test to identify the variance between the structural model of the two different groups. Comparing the chi-square scores can identify the statistically significant difference between two groups ([54]). Finally, we identified the moderating effects on the relationship between independent and dependent variables ([46]).

### 3.4. Entrepreneurship Education at NIT, Tsuruoka College

This study focuses on the entrepreneurship education conducted at the NIT, Tsuroka College. A course called “General Engineering III” is offered to third-year students as part of the four-year engineering program starting from first-year students ([42]). Tsuruoka College was selected for the analysis because it has been providing entrepreneurship education for nine years. Moreover, the merit of its course has been recognized by society, as it has received awards in business contests when collaborating with companies in Tsuruoka City. The first-year course objective is to acquire knowledge of intellectual property (General Engineering I). The second-year course objective is to research firm to create company PR materials through efficiently held group discussions (General Engineering II). The third-year course objective is to work as a team to achieve goals, gain knowledge of entrepreneurs and entrepreneurship, and develop business models while considering the rapidly changing social structure (General Engineering III). The fourth-year course objective is to generate new ideas and create patents (General Engineering IV). The detailed course for third-year course is presented in Table 2.

## 4. Results

### 4.1. Descriptive Analysis

The value of average ranged from 2.01 to 3.96, and value of standard deviation ranged from 0.74 to 0.90, implying that most of the students has high attitude towards entrepreneurship with comparatively lower standard deviation degree. The correlation coefficients suggest there were no high correlation coefficients (Table 3).

### 4.2. Model Fit

We examined the model fit based on CMIN/DF, IFI, CFI, and RMSEA. For the CMIN/DF, the critical value was three or less, indicating that the difference between the hypothesized model and the sample data were within the acceptable range. The CMIN/DF value was 2.719, suggesting that the fit of the hypothesized model was acceptable. An IFI value close to 0.9 or higher is desirable ([25]). The value of 0.9 indicates the fit of the hypothesized model ([52]). A CFI value close to 0.9 or higher is considered desirable, with a value of 0.9 suggesting an acceptable model fit. For RMSEA, the closer the value is to 0.00, the better the fitness ([18]). A value of 0.08 or less is considered desirable. The results of 0.08 suggest a fit of the hypothesized model. Finally, we mitigated the risk of a causal relationship by adopting the common method bias developed by [33] ([33]). A scale unrelated to the constructs in this study was used for analysis. (If a product sold by a company causes serious damage to the environment, I would refuse to buy that product.) A partial correlation analysis was conducted with this as a marker variable. A comparison was made between the controlled and uncontrolled correlation coefficients, and the results showed no common method bias ([27]). The results of convergent validity and estimates are demonstrated in Table 4. All CR values in this study are greater than 0.7, implying a high level of internal consistency in the data. Finally, we have calculated the R-square value for the entrepreneurship intention. The R-square for the entrepreneurial intention variable is 0.407, which indicates that entrepreneurial intention variable can be explained by both attitude and self-efficacy by 40.7%, and the remaining 59.3% is explained by other variables.

### 4.3. Attitude, Subjective-Norm, and Self-Efficacy Toward Entrepreneurship Intention

As described in Figure 2, the results of structural equation modeling showed that attitude and self-efficacy have significant and positive effect on entrepreneurship intention (attitude, β = 0.561, *p* < 0.001; self-efficacy, β = 0.296, *p* < 0.001). Hence, H1 and H1c were supported. In contrary, subjective norms was found to have no significant effect on entrepreneurship intention (β = −0.064, *p* > 0.05), implying that the H1b was not supported.

### 4.4. Entrepreneurship Education Toward Entrepreneurship Intention

To assess whether the entrepreneurship education influenced attitude and self-efficacy on entrepreneurship intention, a multi-group analysis was conducted. As direct effect of subjective norms on entrepreneurship intention found to have negative and insignificant effect; the current study will further continue to discuss on the attitude and self-efficacy. The chi-square difference test showed that the two different groups were statistically and significantly different for attitude and self-efficacy (*p* > 0.001). According to the results of the multi-group analysis, the impact of entrepreneurship education on the relationship between attitude and entrepreneurship intention was found to be positive and significant in both second- and fourth-year students. Similarly, the impact of entrepreneurship education on the relationship between self-efficacy and entrepreneurship intention was found to be positive and significant in both grades. These results indicated that as students take entrepreneurship education, their attitude increases while self-efficacy decreases (Figure 3).

## 5. Discussion

Japan has started its initiatives to develop entrepreneurship intention for youth in junior high and high schools ([59]; [62]), including NIT ([20]). In this context, we aimed to identify the role of entrepreneurship education in developing entrepreneurship intentions specific to engineering majors. The results indicated that entrepreneurship education positively and significantly affects engineering students. The following theoretical and practical implications can support institutions aiming to develop courses to foster entrepreneurial intentions.

### 5.1. Theoretical Implications

The theoretical implication of this study is that it expands the theory of planned behavior by adopting self-efficacy in the research model. This is particularly unique in the engineering field because a literature review found that previous studies have not discussed on theoretical aspect ([41]). Moreover, this study identified different effects of attitude and self-efficacy on entrepreneurship intention among second- and fourth-year students. Previously, many studies adopted this approach to conduct questionnaire surveys right before and after the course finished ([2]; [16]; [13], [14]; [28]; [29]; [35]), which raised the concern of exaggerating the results with the effects of entrepreneurship education ([4]; [15]; [47]). To mitigate this challenge, the current study conducted a questionnaire survey of students one year before and one year after the course. This study is novel as we analyzed the effects of entrepreneurship education rigorously.

The attraction has been on the engineering major students with their technical training to give them power to start developing new technology. Moreover, engineering students have not taken business major course to make them feel pressured and biased in learning and developing entrepreneurship ([55]). Regarding engineering students in entrepreneurship education, [55] ([55]) has conducted a study on university students from UK and France. The quantitative analysis found that attitude, subjective norm, and perceived behavioral control have significantly and positively influenced the intention to become self-employed. Moreover, with the pre- and post-survey on the entrepreneurship education, results found that only subjective norms had significant changes caused by the entrepreneurship education. Comparing the results of this study to current study, there are two theoretical implications. First, the current study has proven that entrepreneurship education has started to develop students’ attitudes and self-efficacy towards entrepreneurship intention. Second, in the current economical circumstance, it is challenging to develop self-efficacy through entrepreneurship education. With the economic challenges faced in the world, this situation is expected to continue. Based on the comparison with the study of [55] ([55]) from the early 2000s, current study has identified the challenges to further analyze the ways to develop subjective norms through incorporating not only students, but also their friends and families through open campus projects to increase the understanding and supports from the society.

### 5.2. Practical Implications

The practical implications are based on the finding that the effect of attitude was higher in the fourth-year students, and self-efficacy was higher in the second-year students. First, studies have suggested that delivering entrepreneurship education focused on developing a business model tends to enhance students’ attitude toward entrepreneurial intention compared to perceived behavioral control ([47]; [56]). Current study results on attitude and self-efficacy have followed these results above, indicating that engineering major students also increase their attitude towards entrepreneurship intention compared to self-efficacy when they receive entrepreneurship education focused on business model development. Second, self-efficacy results align with the findings of a previous study on business students, which states that when students receive more business-related knowledge, they tend to critically evaluate entrepreneurial opportunities ([36]). This study’s results show that this is also relevant for engineering major students. Therefore, when knowledge of entrepreneurship increases, students’ rigorous consideration toward entrepreneurship intention increases, resulting in lower effects on fourth-grade students’ self-efficacy compared to second-grade students. Based on the discussion above, to develop self-efficacy, entrepreneurship education must be well-balanced to develop knowledge, skills, and mindset. Not focusing on either developing business models while creating products, so-called project-based learning, or developing knowledge focusing on a subject such as finance, so-called theory-based learning, but delivering them comprehensively will be the key to fostering students’ entrepreneurship ([5]).

## 6. Conclusions

In this study, we have conducted an analysis to identify factors that influence the entrepreneurship intention for number of institutions which focus on the engineering field for youth by conducting a study at the NIT. With attention paid to the engineering research field in developing students’ entrepreneurship intention, we aim to identify the effects of attitude, subjective norm, and self-efficacy on entrepreneurship intention. Also, to identify the change made through entrepreneurship education comparing the data with the second- and fourth-grade students. Hypotheses 1a and 1c were fulfilled with the results as both attitude and self-efficacy have positively and significantly influenced entrepreneurship intention. However, hypothesis 1b was not fulfilled, as there was no statistical significance in the results. For hypotheses 2a and 2c, we have developed a hypothesis that attitude and self-efficacy will be developed through entrepreneurship education by comparing the data between second and fourth-grade students. The results found that H2a fulfilled our hypothesis; however, H2c did not fulfill our hypothesis, since its value did not increase, but rather decreased after taking the entrepreneurship education. Moreover, H2b did not fulfill our hypothesis by not showing the statistical significance on the results. This conclusion will generate greater value to the engineering research field, researchers, and practitioners who wish to develop practical entrepreneurship education for students to be able to identify the problem and deliver solutions through innovative ideas. Furthermore, this research stream in engineering must shift towards more sophisticated practical entrepreneurship education by developing an incubator for youth to be able to develop products and services through market research while grasping the knowledge from the business field. For engineering students, based on the research study, it is important to deliver both theoretical and practical knowledge to develop their confidence in developing entrepreneurship intention.

### Limitations and Future Research

This study has four limitations. First, we conducted a questionnaire survey and analysis targeting only students from NIT at Tsuroka College. Moreover, the current study did not compare the results of engineering and non-engineering major students. This decreases the validity of identifying engineering students’ characteristics in Japan. Future research can identify the role of entrepreneurship education by comparing students majoring in different subjects to identify the characteristics of students majoring in engineering in developing their entrepreneurship intention. Second, this study focused only on the overall effects of entrepreneurship education delivered at Tsuroka College. Future studies should focus on different types of entrepreneurship education, including theory- and practice-based entrepreneurship education, to identify detailed suggestions for developing entrepreneurship education. Third, due to the protection of students’ information, this study did not clarify the detailed information regarding their entrepreneurial families. In a future study, a chronological study on the graduated students to identify the impact created by taking the entrepreneurship education course with more in depth questions can be expected. Finally, the current study has not targeted the characteristics of Japanese students. However, in a future study, while most of the entrepreneurship education is conducted globally, it is necessary to identify the regional characteristics while comparing the study between nations.

## Figures and Tables

**Figure 1 behavsci-15-00663-f001:**
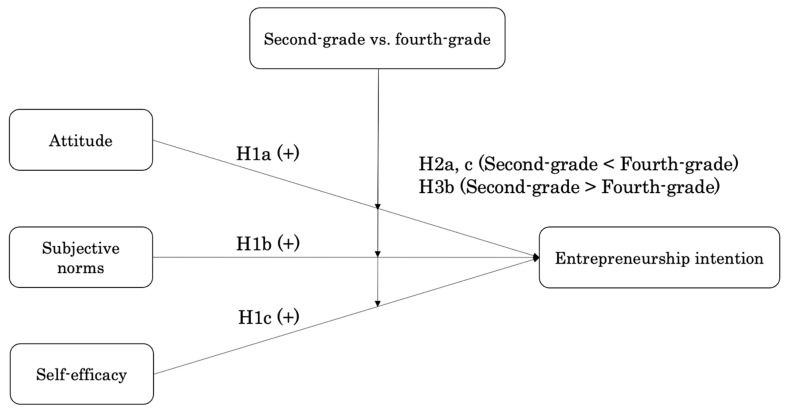
Research model.

**Figure 2 behavsci-15-00663-f002:**
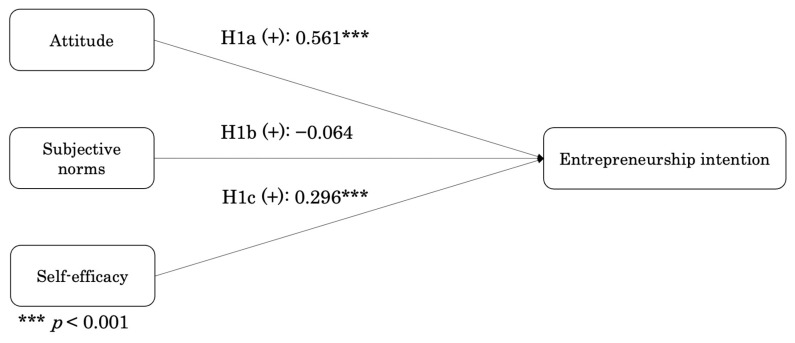
Results of direct effects.

**Figure 3 behavsci-15-00663-f003:**
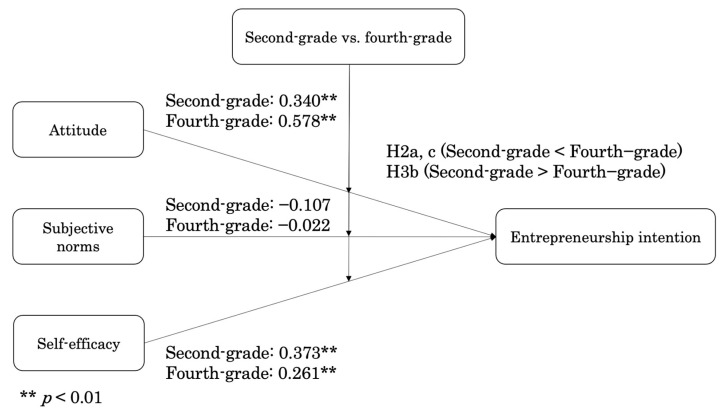
Results of moderating effects.

**Table 1 behavsci-15-00663-t001:** Sample description.

			Second Year (N = 150)		Fourth Year (N = 125)	
Variable		Total	Number	%	Number	%
Gender	Woman	60	36	24.0%	24	19.2%
	Men	213	113	75.3%	100	80.0%
	Others	2	1	0.7%	1	0.8%
Age	16 years old	70	70	46.7%	0	0.0%
	17 years old	76	76	50.7%	0	0.0%
	18 years old	73	3	2.0%	70	56.0%
	19 years old	54	0	0.0%	54	43.2%
	20 years old	0	0	0.0%	0	0.0%
	21 years old	2	1	0.7%	1	0.8%
	22 years old	0	0	0.0%	0	0.0%
Course	Information Systems Engineering	74	39	26.0%	35	28.0%
	Mechanical Engineering	58	36	24.0%	22	17.6%
	Electrical and Electronic Engineering	64	35	23.3%	29	23.2%
	Chemistry and Biology	79	40	26.7%	39	31.2%

**Table 2 behavsci-15-00663-t002:** Course theme for General Engineering III.

Week	Course Content	Weekly Goals
Week 1	Course guidance, information literacy, and information security	Understand the syllabus
Week 2	Guest speaker session 1	Consider and discuss about entrepreneurship
Week 3	Guest speaker session 2	Consider and discuss about entrepreneurship
Week 4	Business model development 1 (team development, idea generation)	Understand the challenge in the society to develop solution
Week 5	Business model development 2 (idea generation and idea organization)	Discuss about business model through an effective group discussion
Week 6	Business model development 3 (customer, key activity, revenue streams)	Develop an implementation plan for business model development
Week 7	Business model development 4 (presentation preparation)	Develop a presentation slides and practice for pitch
Week 8	Presentation day (top eight groups will be invited for general public presentation event)	Discuss the feasibility and improvements for each business model

**Table 3 behavsci-15-00663-t003:** Descriptive analysis.

	Average	Standard Deviation	X1	X2	X3
X1. Attitudes	3.31	0.74			
X2. Subjective norms	3.96	0.76	0.30 **		
X3. Self-efficacy	2.73	0.9	0.41 **	0.21 **	
X4. Entrepreneurship intention	2.01	0.83	0.59 **	0.16 **	0.51 *

Note: * *p* < 0.05, ** *p* < 0.01.

**Table 4 behavsci-15-00663-t004:** Constructs and scale item.

Constructs and Scale Item	Estimates	CR
Attitude ([34])		0.748
Being an entrepreneur implies more advantages than disadvantages to me	0.526	
A career as entrepreneur is attractive for me	0.667	
If I had the opportunity and resources, I’d like to start a firm	0.564	
Being an entrepreneur would entail great satisfaction for me	0.657	
Among various options, I would rather be an entrepreneur	0.744	
Subjective norm ([34])		0.799
If you decided to create a firm, would your family approve of that decision?	0.530	
If you decided to create a firm, would your friends approve of that decision?	0.767	
If you decided to create a firm, would your classmates approve of that decision?	0.821	
If you decided to create a firm, would your teacher at school approve of that decision?	0.677	
Self-efficacy ([17])		0.841
I show great aptitude for creativity	0.598	
I show great aptitude for innovation	0.776	
I show great aptitude for leadership	0.708	
I show great aptitude for problem-solving	0.679	
I can see new market opportunities for new products and services	0.721	
I can develop a working environment that encourages people to try out something new	0.751	
Entrepreneurship intention ([34])		0.880
I am ready to do anything to be an entrepreneur	0.718	
My professional goal is to become an entrepreneur	0.603	
I will make every effort to start and run my own firm	0.562	
I am determined to create a firm in the future	0.932	
I have very seriously thought of starting a firm	0.912	
I have the firm intention to start a firm someday	0.803	

Note: CR = composite reliability.

## Data Availability

Raw data are unavailable to protect students’ personal information at the National Institute of Technology, Tsuruoka College.

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
