# Peer review of "Entrepreneurial Intention Among Engineering Students: The Moderating Role of Entrepreneurship Education in Japan"

_behavsci, 2025, doi:10.3390/bs15050663_

Round 1
Reviewer 1 Report
Comments and Suggestions for Authors
I would like to thank the authors for presenting this manuscript. The structure of the document makes the text easy to read and understand. The steps are clearly shown within each section.
Below, I discuss several issues to improve the manuscript's quality and meet this journal's quality standards. Overall Recommendation: Reconsider after major revisions:
The article addresses an interesting, important and relevant topic. The study aims to contribute to the field, particularly with its approach of extending the theory of planned behaviour by adopting self-efficacy (social cognitive theory) into the research model in the context of student engineers in Japan and evaluating the effects of entrepreneurship education. However, improvements are needed in several areas.
First, the abstract should be revised for clarity and to highlight the study's practical implications.
The literature review is sound, but it would be advisable to incorporate additional studies on entrepreneurship education's effect on the relationship between subjective norms and entrepreneurial intention.
The methodology should be improved. On the one hand, information on participants' consent and anonymity in their questionnaire answers should be indicated. On the other hand, the methodological justification needs to be strengthened, particularly when explaining the choice of structural equations (SEM) and multi-group analysis over other possible methods. Finally, it would be important to examine students' current employment (whether they are currently working, if so, in what capacity and how many years of experience), whether they have previously enjoyed entrepreneurial experiences, whether they come from entrepreneurial families or not. In this sense, it would have been particularly interesting to understand at least the characteristics of the sample in this context.
The results section shows the important findings in an orderly fashion, but it would be advisable to incorporate key information associated with:
- The validation of the sample size.
- What is the explanatory power (R2) of the dependent variable (entrepreneurial intention) that is explained by the model's independent variables?
- The moderating effect size (f2) of entrepreneurial education on the model relationships.
- What is the explanatory power (R2) variation of the dependent variable (entrepreneurial intention) caused by the moderating effect of entrepreneurship education?
The discussion should be developed with a more explicit emphasis on the unique contributions and limitations of the study. One recommendation would be to discuss the results of the variables in concerning previous studies that have examined these variables and reported findings on their relationships. This would provide greater context and depth to the analysis.
The ‘conclusion’ section would be used to reveal how the article adds value to the field of research. However, it is missing from the paper.
As a reader, it is difficult to discern the implications of the findings. Clarifying how the results contribute to the field and addressing their real-world applications would enhance the impact of the paper.
Author Response
Comment 1: First, the abstract should be revised for clarity and to highlight the study's practical implications.
Response 1: Thank you for your comment on the abstract. We have revised the words to improve clarity and added more information on practical implications.
Comment 2: The literature review is sound, but it would be advisable to incorporate additional studies on entrepreneurship education's effect on the relationship between subjective norms and entrepreneurial intention.
Response 2: Thank you for your interesting insights on adding a new study on the entrepreneurship education effect. We have added the Alam et al. (2022) article called "Role of entrepreneurial motivation on entrepreneurial intentions and behaviour: theory of planned behavior extension on engineering students in Pakistan" to improve our literature review. The included sentences are in the following: "Alam et al. (2019) conducted a study with 448 engineering students in Pakistan. This study has analyzed how attitude, subjective norm, and perceived behavioral control influence the entrepreneurial behavior while mediating the entrepreneurial intention. Moreover, the moderating role of entrepreneurial motivation was analyzed. Results found that both attitude and perceived behavioral control have a positive and significant influence on the entrepreneurship intention. However, subjective norm did not significantly influence the entrepreneurship intention. Finally, the mediating role of entrepreneurial intention was found to be significant for attitude and perceived behavior control with entrepreneurial behavior. The course objectives and materials for entrepreneurship education were not clarified in the paper, making it challenging to compare the results with other research papers".
Comment 3: The methodology should be improved. On the one hand, information on participants' consent and anonymity in their questionnaire answers should be indicated.
Response 3: Thank you for your suggestion. Your suggestion is very valuable for our study to be more understandable, especially in the methodology. We have added two sentences at the end of the first section in the methodology to explain the participants' consent and anonymity in the questionnaire survey. The following are the sentences: "The information on participants’ consent was received before the questionnaire survey. Moreover, the anonymity of the questionnaire survey has been protected by only extracting the anonymized data".
Comment 4: The methodology should be improved. On the other hand, the methodological justification needs to be strengthened, particularly when explaining the choice of structural equations (SEM) and multi-group analysis over other possible methods.
Response 4: Thank you for your important advice. For SEM, we have added information in the first section. With the additional source from Hair & Alamer (2022). For Multi-group analysis, we have added new information regarding moderated regression analysis to identify the importance of utilizing multi-group analysis. The following is the new sentences "However, as previous study mentioned that SEM is designed to test the established theoretical frameworks including measurement and structural models and on the other hand, PLS-SEM is designed to identify the user’s structural model objective is to predict and explain the target outcome as obtained by the in-sample and out-of-sample metrics (Hair & Alamer, 2022) and similarly, there is a method called moderated regression analysis to identify the moderator’s effect on the relationship between independent variable and dependent variables. As this study is to compare the different age groups with its aim of identifying the different effects of independent variables on the dependent variables, the current study has selected multi-group analysis".
Comment 5: Finally, it would be important to examine students' current employment (whether they are currently working, if so, in what capacity and how many years of experience), whether they have previously enjoyed entrepreneurial experiences, whethC8:C9er they come from entrepreneurial families or not. In this sense, it would have been particularly interesting to understand at least the characteristics of the sample in this context.
Response 5: Thank you very much for your interesting questions. Currently, the students are still students with no work experience. Moreover, most of the students have not experienced nor heard about entrepreneurship since in Japan, the penetration rate of entrepreneurship is in the early stages. Regarding the entrepreneurial families, it was challenging for us to ask about their families due to the protection of the students' information. In this study, the characteristics of students' information were limited due to the protection of students' information. However, we are planning to conduct a chronological study with these students who have participated in this study, and in the future, we will be able to ask more questions regarding their characteristics. Regarding this issue, we have explained in the limitation section for us to solve in the next study.
Comment 6: • The validation of the sample size.
Response 6: Thank you for your important comment on the target population and its representativeness of the population. Moreover, the confidence level and error have been added to the manuscript in the methodology section. The sentence is in the following: "The total number of valid responses is 275, while the population size is 310. There was a 1.95% sampling error rate for a 95% confidence level. Moreover, the error rate was low enough to be appropriate for a statistical study".
Comment 7: • The moderating effect size (f2) of entrepreneurial education on the model relationships.• What is the explanatory power (R2) variation of the dependent variable (entrepreneurial intention) caused by the moderating effect of entrepreneurship education?
Comment 7: Thank you for your interesting comment. It is very important to understand the moderating effect using moderation analysis. However, this study is to identify the effect of entrepreneurship education through the comparison between changes made before to after taking the entrepreneurship education course. In other words, there is no variable such as entrepreneurship education to multiply with its independent variable to identify the moderating variable's explanatory power. This is due to its necessity to develop a variable that represents entrepreneurship education by asking questions such as "entrepreneurship education was useful". However, this is quite challenging with our research objectives since we have to ask questions to students who have not taken the entrepreneurship education. Based on these circumstances, it is appropriate for us to undertake the multi-group analysis with its chi-square difference tests to identify the effect size and confidence level for each moderating effect on the relationship between the independent variable and the dependent variable.
Comment 8: • What is the explanatory power (R2) of the dependent variable (entrepreneurial intention) that is explained by the model's independent variables?
Response 8: Thank you for your important points. We have calculated the squared multiple correlation, which is equivalent to the explanatory power in the multiple regression analysis, to identify the degree to of the model to predict its outcome. The R-square for the entrepreneurial intention variable is 0.407, which indicates that the entrepreneurial intention variable can be explained by both attitude and self-efficacy by 40.7%, and the remaining 59.3% is explained by other variables. We have included this data in the manuscript as "Finally, we have calculated the R-squared value for the entrepreneurship intention. The R-square for the entrepreneurial intention variable is 0.407, which indicates that the entrepreneurial intention variable can be explained by both attitude and self-efficacy by 40.7%, and the remaining 59.3% is explained by other variables.
Comment 9: The discussion should be developed with a more explicit emphasis on the unique contributions and limitations of the study. One recommendation would be to discuss the results of the variables in concerning previous studies that have examined these variables and reported findings on their relationships. This would provide greater context and depth to the analysis.
Response 9: Thank you very much for your important comment. We have added theoretical implications. The included sentences are in the following: " First, the current study has proven that entrepreneurship education has started to develop students’ attitudes and self-efficacy towards entrepreneurship intention. Second, in the current economic circumstances, it is challenging to develop self-efficacy through entrepreneurship education. With the economic challenges faced in the world, this situation is expected to continue. Based on the comparison between the study from the early 2000s and the current study has identified the challenge has been identified to further analyze the ways to develop subjective norms through incorporating not only students but also their friends and families through open campus projects to increase the understanding and support from society.
Comment 10: The ‘conclusion’ section would be used to reveal how the article adds value to the field of research. However, it is missing from the paper.
As a reader, it is difficult to discern the implications of the findings. Clarifying how the results contribute to the field and addressing their real-world applications would enhance the impact of the paper.
Response: Thank you for your important points. We have added a sentence in conclusion that explains the results' contribution to the field and addresses the real-world applications. The sentences are in the following: "This conclusion will generate greater value to the engineering research field, researchers, and practitioners who wish to develop practical entrepreneurship education for students to be able to identify the problem and deliver solutions through innovative ideas. Furthermore, this research stream in engineering must shift towards more sophisticated practical entrepreneurship education by developing an incubator for youth to be able to develop products and services through market research while grasping the knowledge from the business field. For engineering students, based on the research study, it is important to deliver both theoretical and practical knowledge to develop their confidence in developing entrepreneurship intention".
Reviewer 2 Report
Comments and Suggestions for Authors
This manuscript offers a valuable and timely contribution to the field of entrepreneurship education by extending the Theory of Planned Behaviour through the inclusion of self-efficacy, particularly within the relatively underexplored context of engineering education in Japan. The longitudinal design is a clear strength, adding robustness to the findings, and the comparison between second- and fourth-year students yields practical insights that could inform curriculum development. The paper’s theoretical contribution is promising; however, it would benefit from a more clearly articulated rationale for its novelty and how the study design addresses limitations found in previous research. I encourage the authors to revise the manuscript for clarity, academic tone, and coherence, particularly in relation to theoretical framing and the connection between findings and implications. With these improvements, the manuscript has a strong potential to make a meaningful impact in the field.
Author Response
Comment 1: It would benefit from a more clearly articulated rationale for its novelty and how the study design addresses limitations found in previous research.
Response 1: Thank you for your comment. We have added new information at the end of the literature review. The sentences are in the following: "Moreover, the novelty of this study is that the current study has adopted a new methodology to analyze the effect of entrepreneurship education by comparing the data between second- and fourth-year students while conducting a questionnaire survey of 310 students at the same time. This novelty has minimized the risk of gathering exaggerated results on the effect of attitude, subjective norm, and self-efficacy on entrepreneurship intention. Moreover, to conduct a study based on the theory of planned behavior while improving this theory by changing the perceived behavioral control to self-efficacy while combining the social cognitive theory, this study has created a new pathway to analyze the entrepreneurship education for engineering major students.
Comment 2: I encourage the authors to revise the manuscript for clarity, academic tone, and coherence, particularly about theoretical framing and the connection between findings and implications.
Response 2: Thank you very much for your important comment. We have added theoretical implications. The included sentences are in the following: " First, the current study has proven that entrepreneurship education has started to develop students’ attitudes and self-efficacy towards entrepreneurship intention. Second, in the current economic circumstances, it is challenging to develop self-efficacy through entrepreneurship education. With the economic challenges faced in the world, this situation is expected to continue. Based on the comparison between the study from the early 2000s and the current study has identified the challenge has been identified as to further analyze the ways to develop subjective norms through incorporating not only students but also their friends and families through open campus projects to increase the understanding and support from society.
Reviewer 3 Report
Comments and Suggestions for Authors
Comments and Suggestions for Authors
The authors must clearly indicate in the introduction the objective of the work presented
Literature reviewed (section 2) is scarce. It should be added to the introduction or section 3, as the author sees fit.
The authors comment that there are few studies on the topic. Perhaps they are scarce in Japan, and further research is needed, but not so in the rest of the world. Google Scholar contains several interesting analyses that can help to identify differences with other regions.
Hypothesis Development (section 3) is well-thought-out. The authors analyze a considerable amount of literature, and the literature is up-to-date.
In Methodology (section 4), more information should be included: What is the target population? Is the sample representative of that population? If so, what is its confidence level and error?
Results (section 5) are well presented
In Discussion (section 6), you should include the conclusions. The authors should indicate whether the objective was met and whether the hypotheses were fulfilled.
Author Response
Comment 1: The authors must indicate in the introduction the objective of the work presented.
Response 1: Thank you for your important suggestion. As you suggested, we have added sentences to explain the objectives of this study. The sentences are in the following: "The objective of this study is to conduct a quantitative analysis on the engineering major students while identifying the role of entrepreneurship education. We will conduct analysis and discussion while combining the theory of planned behavior and social cognitive theory. Specifically, we will identify how attitude, subjective norms, and self-efficacy will influence entrepreneurship intention. Also, how entrepreneurship education will moderate each relationship".
Comment 2: The Literature reviewed (section 2) is scarce. It should be added to the introduction or section 3, as the author sees fit.
Response 3: Thank you for your interesting insights on adding a new study on the entrepreneurship education effect. We have added the Alam et al. (2022) article called "Role of entrepreneurial motivation on entrepreneurial intentions and behaviour: theory of planned behavior extension on engineering students in Pakistan" to improve our literature review. The included sentences are in the following: "Alam et al. (2019) conducted a study with 448 engineering students in Pakistan. This study has analyzed how attitude, subjective norm, and perceived behavioral control influence the entrepreneurial behavior while mediating the entrepreneurial intention. Moreover, the moderating role of entrepreneurial motivation was analyzed. Results found that both attitude and perceived behavioral control have a positive and significant influence on the entrepreneurship intention. However, subjective norm did not significantly influence the entrepreneurship intention. Finally, the mediating role of entrepreneurial intention was found to be significant for attitude and perceived behavior control with entrepreneurial behavior. The course objectives and materials for entrepreneurship education were not clarified in the paper, making it challenging to compare the results with other research papers. Moreover, we have added a literature review to the introduction.
Comment 3: The authors comment that there are few studies on the topic. Perhaps they are scarce in Japan, and further research is needed, but not so in the rest of the world. Google Scholar contains several interesting analyses that can help identify differences with other regions.
Response 3: Thank you for your insightful comment. We have checked Google Scholar for more articles which is related to this current study and added them to the literature review. Regarding the difference between other regions, we are also interested in comparing how entrepreneurship education influences students in different countries. Therefore, we have added this challenge in the limitation section for prospects. The sentences are in the following: " Finally, the current study has not targeted the characteristics of Japanese students. However, in the future study, while most of the entrepreneurship education is conducted globally, it is necessary to identify the regional characteristics while comparing the study between nations".
Comment 4: In Methodology (section 4), more information should be included: What is the target population? Is the sample representative of that population? If so, what is its confidence level and error?
Response 4: Thank you for your important comment on the target population and its representativeness of the population. Moreover, the confidence level and error have been added to the manuscript in the methodology section. The sentence is in the following: "The total number of valid responses is 275, while the population size is 310. There was a 1.95% sampling error rate for a 95% confidence level. Moreover, the error rate was low enough to be appropriate for a statistical study".
Comment 5: In Discussion (section 6), you should include the conclusions. The authors should indicate whether the objective was met and whether the hypotheses were fulfilled.
Response 5: Thank you for your suggestion. Combining the other reviewer's comments, we have created a conclusion section to explain whether the objectives were met and the hypotheses were fulfilled. The following is the included sentence: "In this study, we have analyzed to identify factors which influence the entrepreneurship intention for several institutions that focus on the engineering field for youth by conducting a study at the NIT. With attention paid to the engineering research field in developing students’ entrepreneurship intention, we aim to identify the effects of attitude, subjective norm, and self-efficacy on entrepreneurship intention. Also, to identify the change made through entrepreneurship education comparing the data with the second and fourth-grade students. Hypotheses 1a and 1c were fulfilled with the results, as both attitude and self-efficacy have positively and significantly influenced entrepreneurship intention. However, hypothesis 1b was not fulfilled, with no statistical significance in the results. For hypotheses 2a and 2c, we have developed a hypothesis that attitude and self-efficacy will be developed through entrepreneurship education by comparing the data between second and fourth-grade students. The results found that H2a fulfilled our hypothesis, however, H2c did not fulfill our hypothesis since it did not increase but rather decreased its value after taking the entrepreneurship education. Moreover, H2b did not fulfill our hypothesis by not showing the statistical significance of the results".